# When to stop: Understanding the landscape of extreme-duration cardiopulmonary resuscitation practices among pediatricians in Sudan

**Mohammed Abdulrahman Alhassan** [ID]*

Department of Pediatrics, College of Medicine, Prince Sattam Bin Abdulaziz University, Alkharj, Riyadh, Saudi Arabia

* mhmdarhafeez@yahoo.com, ma.alhassan@psau.edu.sa

## Abstract

### Background

Pediatric cardiopulmonary resuscitation (CPR) is a life-saving intervention, but its effectiveness in extreme durations remains debated. This study aimed to explore the frequency and decision-making regarding prolonged CPR (PCPR) practices in hospitalized Sudanese children.

### Methods

A web-based cross-sectional survey was conducted among pediatricians and pediatric trainees in Sudan. The survey investigated their experience with prolonged, ultra-prolonged, and extreme-duration CPR and factors influencing termination decisions.

### Results

Ninety-six Sudanese pediatricians and trainees responded to the survey, reporting varied experiences with prolonged CPR durations: over half (51%) and 81% of respondents reported encountering extreme-duration (> 12 hours) and ultra-prolonged (> 2 hours) CPR at least once, respectively. Around 5% and 1% reported to have encountered CPR durations of 48–72 hours and more than 72 hours, respectively, while 18% reported a 12–14-hour as their record high CPR duration. Four participants reported encountering extreme-duration (> 12 hours) CPR more than 15 times. Respondents most frequently (41%) cited the absence of a pulse and heartbeat as the primary factor for terminating CPR. A vast majority acknowledged ethical considerations (84%) and a lack of clear protocols (83%) as a barrier to terminating PCPR. Thematic analysis of an open question revealed a critical need for a standardized protocol addressing PCPR, enhanced CPR training, and better post-resuscitation support.

**Data availability statement:** All relevant data are within the paper and its Supporting Information files.

**Funding:** The authors extend their appreciation to Prince Sattam bin Abdulaziz University for funding this research work through the project number (PSAU/2024/03/29191).

**Competing interests:** The authors have declared that no competing interests exist.

## Conclusions

This study revealed a relatively high frequency of extended-duration CPR in Sudanese pediatric settings. Termination decisions focused on cardiac activity becoming clinically undetectable rather than brain death signs or specific cutoff CPR durations. The absence of a clear protocol on when to terminate CPR seems to contribute the most to this phenomenon. Further research on patient outcomes after prolonged CPR in this context is warranted.

## Introduction

Prolonged cardiopulmonary resuscitation (CPR) in children is not uniformly defined across the literature, but it generally refers to resuscitation efforts that extend beyond the typical duration expected to achieve return of spontaneous circulation. Durations exceeding 20–30 minutes were often considered prolonged in previous studies [1,2]. While prolonged CPR in children is often associated with poor outcomes [3], there are reported incidences with favorable neurological prognosis and survival in adults, particularly in specific circumstances or patient populations [4]. However, the author could not find published studies addressing the outcomes of extended-period CPR (e.g., several hours or longer) in children.

In contrast to adults, where primary cardiac dysrhythmia is often the cause of cardiac arrest, children typically experience cardiac arrest following a respiratory collapse [5]. Respiratory arrest is thus more commonly a precursor to cardiac arrest in pediatric patients. Respiratory arrest in pediatric patients, while maintaining a palpable pulse or audible heartbeat, is a recognized clinical scenario during cardiopulmonary resuscitation (CPR) in pediatric hospital practice [6]. In settings lacking mechanical ventilation, such events may necessitate extended periods of manual bag-valve-mask ventilation (BMV) [7].

In Sudan, where most pediatric hospitals lack mechanical ventilators and monitors, sessions of ultra-prolonged BMV or CPR are not uncommon. These labor-intensive and resource-demanding endeavors are usually terminated only when pulses and heartbeats become undetectable, and the pupils become fixed and dilated. Otherwise, these sessions can last for several hours or even a few days. Typically, junior members of the medical team take turns manually ventilating the child, often interspersed with periods of cardiac compressions.

The outcomes of prolonged CPR and the decisions surrounding when (and if) to discontinue CPR attempts remain subjects of further study, as well as academic and legislative debate. In high-resource settings, prolonged CPR in children is fraught with medical, ethical, and legal controversy. Studies show that extended resuscitation is usually associated with poor outcomes in pediatric cardiac arrest [8], yet deciding when to stop is extremely difficult. Ethically and legally there is strong pressure to "do everything" for a child – in fact, many jurisdictions exclude pediatrics from termination-of-resuscitation protocols, meaning providers must typically continue CPR until hospital arrival unless death is certain [9]. This reluctance to cease efforts

stems from outcome uncertainty: rare cases of survival after lengthy CPR make it hard to set firm cutoffs, so guidelines often emphasize case-by-case judgment and even advise involving the family in termination decisions once resuscitation has exceeded ~30 minutes with no response [9]. In low-resource settings such as Sudan, these debates take on an additional significance as limited staffing, infrastructure, and supplies add a pragmatic layer. Local pediatric guidelines acknowledge that even basic equipment and drugs may be unavailable – standard CPR protocols have to be modified to fit reality [10]. Consequently, discussions focus on how to balance ideal care with feasibility, and international experts have highlighted the need for context-specific resuscitation protocols, noting that many "international" CPR guidelines (designed for high-resource contexts) are often not applicable in resource-limited environments [11].

Consequently, it is not surprising that a clear protocol restricting or informing the duration of CPR in children in Sudan is lacking. In this study, the author aims to investigate the pediatricians' experiences with prolonged (> 30 minutes- PCPR), ultra-prolonged (> 2 hours- UPCPR), and extreme-duration (>12 hours- EDCPR) BMV or CPR. Given the poor documentation and difficulty in accessing older records, the author chose to gather data through surveys capturing pediatricians' experiences instead of reviewing case logs.

## Materials and methods

### Study design

This study employed a cross-sectional survey design to explore the experience of Sudanese pediatricians and pediatric trainees on prolonged (> 30 minutes), ultra-prolonged (> 2 hours), and extreme-duration (> 12 hours) CPR.

### Setting, participants, and sampling

The target population included all Sudanese consultants/specialists and pediatric residents with a minimum of one year of recent hospital experience in Sudan. According to a 2010 report, Sudan had only 212 qualified pediatricians—with 112 being female—registered by that time [12]. There's no more recent official national count available in the public domain. Convenience sampling was employed due to the difficulty of obtaining a comprehensive national sampling frame for pediatricians. The representativeness of the sample was evaluated by comparing the demographic characteristics of the survey respondents with those of the broader population of Sudanese pediatric professionals. The recruitment strategy involved two main channels:

1. Professional Pediatric WhatsApp Groups: The researcher identified seven inclusive and active professional WhatsApp groups specifically for Sudanese pediatricians and pediatric trainees. A study information sheet and the web-based questionnaire link were disseminated to all group members.

2. Personal Professional Contacts: The study author leveraged his professional network to distribute the questionnaire link directly to colleagues who fit the study criteria.

### Questionnaire and data collection

A web-based questionnaire was administered using Google Forms. The survey was available for completion over a nine-day period (June 26th to July 4th, 2024), with daily reminders sent via WhatsApp to encourage participation. The questionnaire included two filtering questions, ten closed-ended items, and one open-ended item. It was developed by the author based on relevant literature, personal insights, and expert opinion to capture information on the frequency and circumstances of prolonged CPR episodes, decision-making processes, and barriers to terminating CPR.

To accommodate participants' linguistic preferences, the questionnaire was prepared in both English and Arabic. A forward–backward translation process was used to ensure linguistic and conceptual consistency. Both versions were pilot-tested with five pediatricians, and only minor wording adjustments were made to enhance clarity.

Since the questionnaire did not involve scales or psychometric constructs, formal construct or criterion validity were not required. However, the questionnaire underwent expert review to ensure face and content validity. Feedback was obtained from pediatricians familiar with CPR practices to confirm that the items were clear, relevant, and aligned with the study objectives. Minor modifications were made accordingly.

## Thematic analysis

Open-ended responses were analyzed using thematic analysis. The author followed Braun and Clarke's six-phase approach [13], beginning with familiarization with the data and generation of initial codes. These codes were then organized into overarching themes. Coding was conducted manually by the author, and emerging categories were refined iteratively. Responses in both English and Arabic were analyzed together after forward–backward translation. The final themes were derived inductively from the data and reflected recurring patterns of opinion and experience regarding prolonged CPR in pediatric settings. Minor language corrections were made to improve clarity without altering the original meaning.

## Data analysis

Descriptive statistics were used to summarize the closed-ended responses. Frequencies and percentages were reported for categorical data. For one continuous variable, the median and range were calculated based on non-normality assumption. Open-ended responses were analyzed thematically using a qualitative content analysis approach. A colleague pediatrician helped in cross-checking the themes derived from the reposes. All analyses were conducted using STATA (StataCorp. 2021. Stata Statistical Software: Release 17). Figures were created with ChatGPT (OpenAI 2024).

## Ethical considerations

The study protocol was reviewed and approved by The Standing Committee of Bioethics Research at Prince Sattam Bin Abdulaziz University (approval No. SCBR-303/2024). Participants were informed about the study's purpose, the voluntary nature of their participation, and the anonymity and confidentiality of their responses before completing the questionnaire. A consent statement was included in the questionnaire.

## Results

Ninety-six participants responded by completing the questionnaire. Of these, 62 (65%) were pediatric residents and medical officers, and 34 (35%) were consultant and specialist pediatricians. Most of the residents and medical officers had experience in pediatrics ranging from 3 to 5 years, while consultants and specialists had experience of 5–15 years. Seventy-six respondents (79%) reported a lack of mechanical ventilators in the hospitals in which they currently work (Table 1).

In response to the question "How long was the longest CPR (or bag and mask ventilation) session you or your unit have been involved in with a pediatric patient, excluding neonates?", the most frequently reported durations were 2–3 hours (n = 18; 19%) and 12–14 hours (n = 17; 18%). Notably, durations from 30 to 42 hours were also commonly reported, constituting 14% of the responses. Prolonged resuscitation beyond 24 hours was less common, yet there were reports of 48–72 hours (n = 5; 5.21%) and even exceeding 72 hours (n = 1; 1.04%) (Fig 1).

The author analyzed the responses to the question "How many times did you encounter situations where CPR or BMV were continued for more than 30 minutes, 2 hours, and 12 hours in a child you or your unit care for (excluding neonates)?". While 49% of the respondents reported having never encountered CPR/BMV extending beyond 12 hours, 51% reported at least one such encounter. Four participants (4.2%) reported experiencing such extreme-duration CPR/BMV events more than 10 times (Table 1 and Fig 2). For ultra-prolonged CPR events (> 2 hours), most participants (81%) reported to have encountered one or more such events. The vast majority (97%) have experienced at least one prolonged

**Table 1. Participants' responses.**

| Character/response | Number | Percentage |
|---|---|---|
| **Experience: residents/SHOs** | | |
| 1 to 2 years | 8 | 12.90 |
| 2 to 3 years | 12 | 19.35 |
| 3 to 4 years | 15 | 24.19 |
| 4 to 5 years | 18 | 29.03 |
| 5 to 6 years | 5 | 8.06 |
| 6 to 7 years | 2 | 3.23 |
| > 7 years | 2 | 3.23 |
| **Experience: Consultants/specialists** | | |
| Less than 5 years | 8 | 23.53 |
| 5 to 15 years | 22 | 64.71 |
| 16 to 25 years | 3 | 8.82 |
| More than 35 years | 1 | 2.94 |
| Less than 5 years | 8 | 23.53 |
| **Availability of mechanical ventilators in your hospital** | | |
| Not available | 76 | 79.17% |
| Available | 20 | 20.83% |
| **Encounters of PCPR** | | |
| 0 times | 3 | 3.13% |
| 1-5 times | 57 | 59.38% |
| 6-10 times | 15 | 15.63% |
| 11-20 times | 14 | 14.58% |
| 21+ times | 7 | 7.29% |
| **Encounters of UPCPR** | | |
| 0 times | 18 | 18.75% |
| 1-5 times | 57 | 59.38% |
| 6-10 times | 13 | 13.54% |
| 11-20 times | 6 | 6.25% |
| 21+ times | 2 | 2.08% |
| **Encounters of EDCPR** | | |
| 0 times | 47 | 48.96% |
| 1-5 times | 44 | 45.83% |
| 6-10 times | 1 | 1.04% |
| 11-20 times | 3 | 3.12% |
| 21+ times | 1 | 1.04% |
| **Single most important factor when deciding to terminate resuscitation efforts** | | |
| Disappearance of heartbeat and/or pulse | 39 | 41.49% |
| Brain death signs (despite present heartbeat or pulse) | 26 | 27.66% |
| Duration of the CPR effort without return of spontaneous breathing | 14 | 14.89% |
| Consultant's or specialist's decision | 8 | 8.51% |
| Pupil size and reaction (despite present heartbeat or pulse) | 4 | 4.26% |
| Parent/caregiver wishes to stop resuscitation | 1 | 1.06% |
| Pupil size and reaction plus disappearance of heartbeat | 1 | 1.06% |
| Other | 1 | 1.06% |

*(Continued)*

**Table 1.** (Continued)

| Character/response | Number | Percentage |
|---|---|---|
| **Barriers to terminating PCPR, UPRPC, and EDCPR[#]** | | |
| Ethical considerations (concerns about the ethical implications of stopping CPR) | 81 | 84.38% |
| Lack of clear protocols for termination of CPR | 80 | 83.33% |
| Uncertainty about the patient's potential for recovery | 63 | 65.62% |
| Fear of legal consequences if CPR is stopped too early | 51 | 53.12% |
| Insufficient communication or support from consultants | 42 | 43.75% |
| **Who conducts CPR when it extends beyond 2 hours?[*]** | | |
| Residents: often | 57 | 63.33 |
| Interns: often | 50 | 55.56 |
| Senior house officers: often | 35 | 43.21 |
| Nursing staff | 28 | 37.33 |
| Consultants/specialists: never/rarely | 65 | 82.28 |

[#] Multiple responses allowed.

[*] Only modes are reported. Not all participants responded (optional question).

SHO: senior house officer; PCPR, UPCPR, EDCPR: prolonged, ultra-prolonged, extreme-duration cardiopulmonary resuscitation.

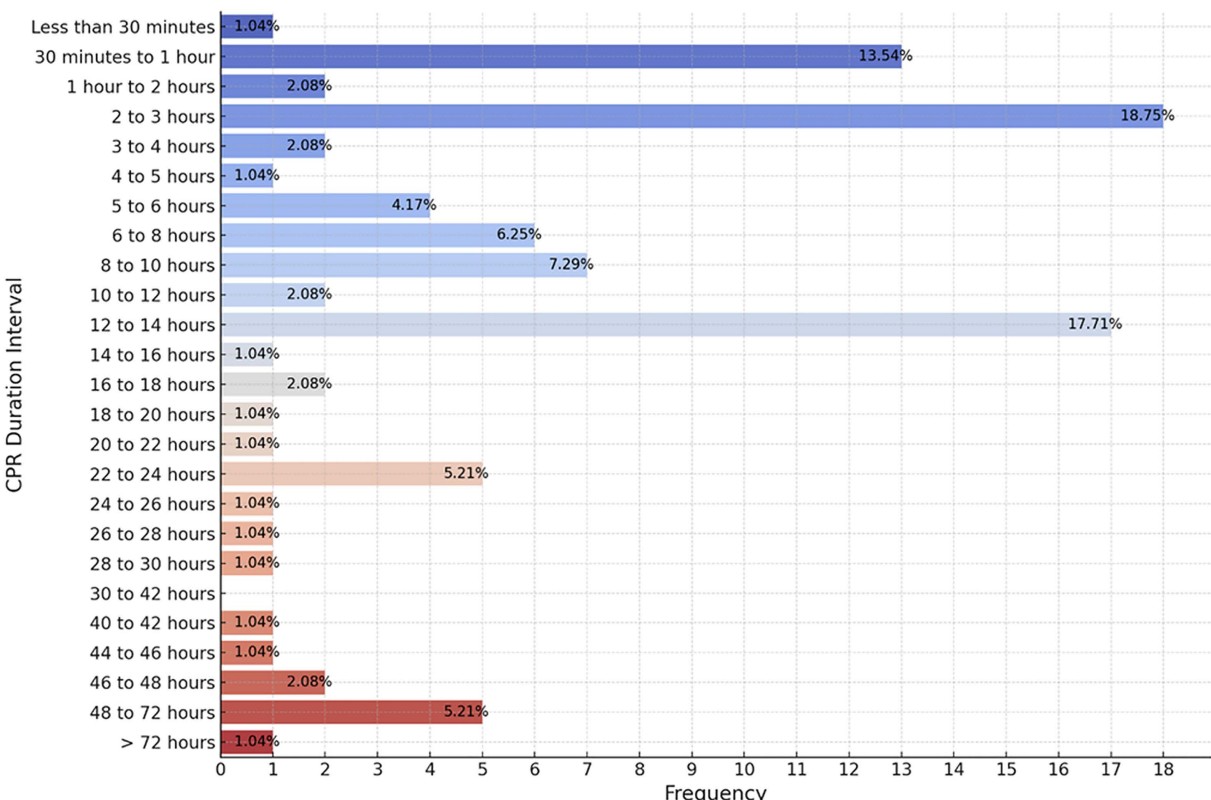

**Fig 1. The frequency of longest reported cardiopulmonary resuscitation (CPR) duration experienced (excluding neonates).**

(> 30 minutes) CPR event. Twenty-one (22%) participants have encountered prolonged CPR/BMV more than 10 times. The maximum number of encounters reported for EDCRP, UPCRP, and PCRP were 21, 26, and 31, respectively (Table 1 and Fig 2).

Participants were asked whether they could recall any instances in their experience where a pediatric patient who received UPCPR and EDCPR survived until hospital discharge (excluding neonates); 18 (19%) and 12 (13%) replied positively, respectively. The most frequently cited factor for deciding to terminate resuscitation efforts in case of CPR/BMV for more than 2 hours is the disappearance of heartbeat and/or pulse, noted by 39 (41%) respondents. Brain death signs, despite the presence of a heartbeat or pulse, are the second most common factor, indicated by 28% of the respondents. A majority of respondents identified the lack of clear protocols (83%) and ethical considerations (84%) as major barriers to stopping CPR/BMV in pediatric patients with prolonged respiratory arrest and preserved heartbeat or pulse. House officers (interns) are more likely to perform the UPCPR and EPCPR, followed by medical officers. Specialists and nursing staff seldom conduct prolonged CPR (Table 1).

### Thematic analysis

In this study, the author explored healthcare professionals' perspectives on prolonged cardiopulmonary resuscitation (CPR) in hospitalized children through the questions "Please share any comments, insights, or experiences you have regarding the prolonged CPR in pediatric patients with respiratory arrest and preserved pulse or heartbeat. This

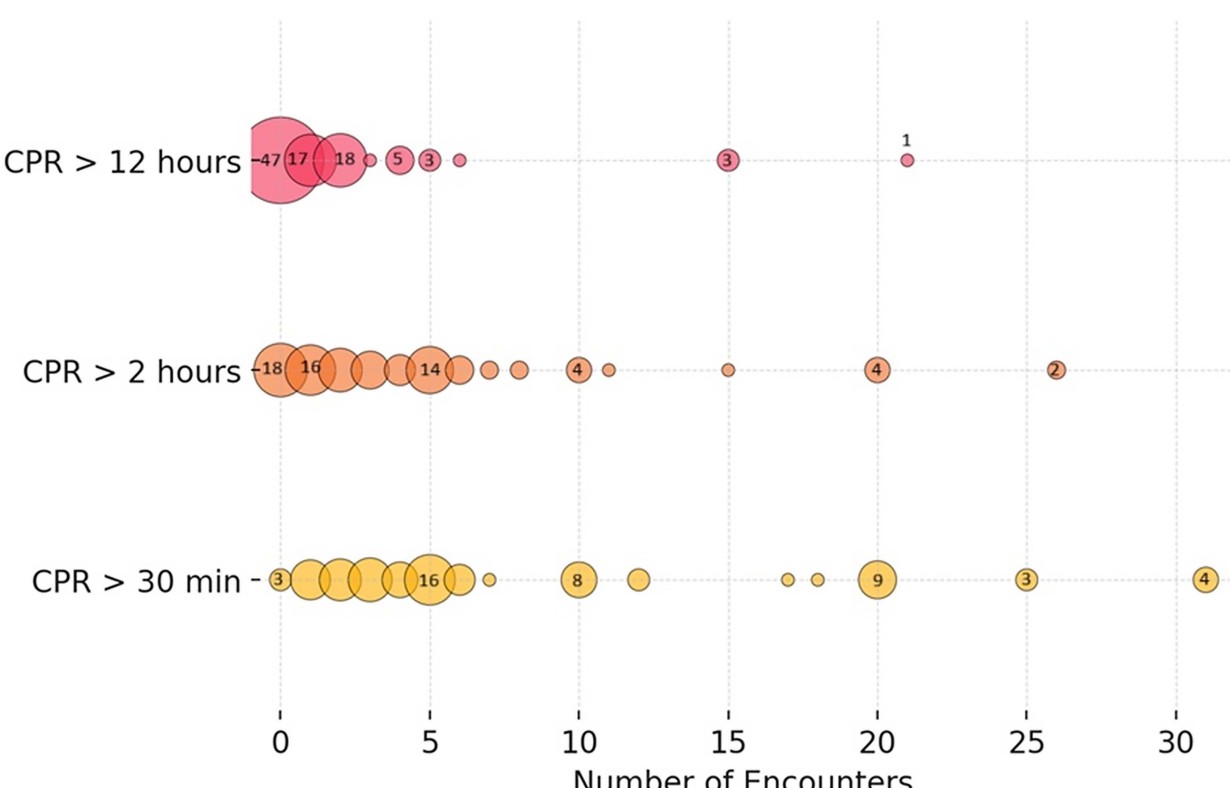

**Fig 2. Responses to the question: "How many times did you encounter situations where Cardiopulmonary resuscitation or bag-mask-valve ventilation were continued for more than 30 minutes, 2 hours, and 12 hours in a child you or your unit care for (excluding neonates)?".** The size of the bubble corresponds to the relative frequency at which each number was mentioned.

**Table 2. Thematic analysis of the open-ended question#.**

| Theme | Count | Percentage (%) |
|---|---|---|
| Critical need for developing a clear protocol/policy for (when to stop) prolonged CPR/BMV (or when to continue prolonged CPR in the absence of MV) | 36 | 28.57 |
| Need for mechanical ventilator (unavailability makes prolonged CPR futile) | 21 | 16.67 |
| Need for more/continuous training on CPR | 12 | 9.52 |
| Lack of proper post-resuscitation critical care (need for pediatric intensive care setting) | 11 | 8.73 |
| Prolonged CPR/BMV is futile/has poor outcome | 7 | 5.55 |
| Shortage/lack of essential medicines and equipment/resources to undertake prolonged CPR. | 6 | 4.76 |
| Need for a dedicated resuscitation team in each hospital | 6 | 4.76 |
| Difficult decision-making around prolonged CPR | 5 | 3.97 |
| Policy/decision-making should accommodate religious and legal aspects. | 5 | 3.97 |
| Decisions around prolonged CPR should be individualized. | 4 | 3.17 |
| Emotional toll/moral burden on doctors if CPR is stopped while heart is still beating | 3 | 2.38 |
| Other themes reported by one or two participants | 10 | 7.94 |
| Total | 126 | |

# Each participant's response could contribute more than one theme. CPR: cardiopulmonary resuscitation; BMV: bag-valve-mask ventilation; MV: mechanical ventilation.

could include your thoughts on decision-making processes, ethical considerations, impacts on the healthcare team, or any other aspect you find significant." Participants provided insights that were categorized into several thematic areas (Table 2 and S1 File).

**Critical need for protocols and policies.** The most frequently cited theme, mentioned in 28.6% of responses, was the critical need for developing clear protocols and policies around prolonged CPR. One respondent emphasized, "In Sudan, we don't have clear guidelines or protocols when to stop prolonged CPR. We usually face such cases with no mechanical ventilation facilities. Decisions usually are individual and depend on the acting team. In some situations, staff exhaustion can be one of the causes of stopping such prolonged CPR. Ethical consideration is one of the most (important) factors to continue such events." Another added, "There should be a clear protocol that we follow, such that we take a specific period of time and even if the heartbeat continues, we stop. Consultants must talk to the family and explain to them."

**Mechanical ventilator availability.** A significant portion of participants (16.7%) highlighted the need for mechanical ventilators. One participant noted, "No practical benefit of prolonged CPR beyond 30 minutes if there are no mechanical ventilatory support facilities and ICU access." Another stated, "(We conducted) prolonged bag and mask ventilation. Due to lack of mechanical ventilation and proper post-resuscitation care, the patient (developed respiratory arrest) again, and CPR was conducted again but unfortunately, we lost the patient."

**Continuous training on CPR.** The need for continuous training on CPR (and life support) was another important theme, identified in 9.5% of responses. One participant highlighted, "My advice is to make sure that all healthcare providers are able to deliver efficient CPR."

**Post-resuscitation critical care support.** Approximately 8.7% of participants recognized the importance of post-resuscitation critical care support. One respondent shared, "In our practice, we conduct CPR properly to save the patient. If CPR is started early, and the patient is lucky enough (to restore respiratory effort), then we are stuck in searching for

an ICU bed and mechanical ventilator to complete the process with post-resuscitation care. We do our best with what we get."

**Additional challenges.** Other themes included:

- Futility of prolonged CPR: Approximately 5.6% of participants emphasized that prolonged CPR or bag-mask ventilation (BMV) is futile or has poor outcomes. A participant summed it up, "Let them lay in peace."

- Need for dedicated resuscitation teams: A need for specialized resuscitation teams in each hospital was noted by 4.8% of participants. A quote from one participant reads: "Because we would literally be standing on the arrested patient (performing CPR/bag-mask-valve ventilation) all day long, the rest of the patients would be neglected."

- Shortage of essential resources: Several participants highlighted the shortage/lack of basic and essential medicines and equipment to undertake prolonged CPR. Oxygen and adrenaline were specifically mentioned.

- Decision-making challenges: About 4% of participants brought up the difficulty of decision-making around prolonged CPR. One participant mentioned, "For me, there is no benefit from prolonged cardiopulmonary resuscitation for children, as it is stressful for the medical staff, especially since its termination depends on the decision from the (senior doctor), who is usually reluctant to make this decision for a period exceeding 12 hours for fear of legal problems."

- Accommodating religious and legal aspects: Another 4% of participants highlighted the need to accommodate religious and legal considerations in prolonged CPR. One respondent explained, "I believe that an opinion or Fatwa (formal ruling or interpretation of Islamic law) must be taken from scholars and religious scholars because in many cases this is a fundamental concern in the decision to stop resuscitation."

- Individualized decisions: Some participants (3.2%) emphasized the point that decisions around prolonged CPR should be individualized.

- Emotional toll and moral burden: About 2.4% of participants highlighted the emotional toll and moral burden on doctors during prolonged CPR. One participant reflected, "Ethically and emotionally, you cannot stop CPR while the patient's heart is still beating." Another one wrote: "A feeling of guilt that if you had continued the resuscitation process, he could have returned to life, and you (wouldn't have felt) responsible for a mother losing her child."

## Discussion

The survey findings revealed that a significant proportion of Sudanese pediatricians and trainees (51%) have encountered extreme-duration CPR (lasting > 12 hours) at least once in their practice. Furthermore, 81% reported encountering incidents of UPCPR (CPR > 2 hours) at least once, with 22% experiencing such an event more than six times. A purposeful literature search indicates that the longest reported durations of CPR attempts are 291 minutes in children and 344 minutes in adults [14,15]. Our survey results thus highlight a distinct and unique landscape where even more extreme durations of CPR are a common practice in Sudanese pediatric care.

Such extreme durations may reflect an underlying emphasis on the preservation of life, regardless of potentially unfavorable or adverse neurological outcomes. While pediatric CPR for more than 20 minutes has historically been considered futile, this notion has been challenged by recent findings. Matos et al. (2013) report that performing CPR for longer than 20 minutes is not necessarily futile in certain patient illness categories. The link between the duration of CPR and neurological outcomes is even more obscure in the literature than that between the duration of CPR and survival outcomes, further complicating the decision-making process regarding the termination of prolonged CPR [2,16,17]. All the reviewed studies addressing the outcomes of prolonged CPR in children, however, did not consider CPR durations as extreme as those in our study, and thus their applicability to our situation may not be accurate.

Prolonging CPR attempts to such extreme durations most likely stems from clinical scenarios where a patient develops respiratory arrest but retains a pulse or heartbeat, particularly in settings lacking continuous cardiac monitoring and mechanical ventilation. The use of cessation of pulse or heartbeat as the single most common factor governing termination decisions (41%) supports this hypothesis. Only 28% and 15% of participants used brain death signs and a cutoff CPR duration, respectively, to terminate prolonged CPR. This may mirror the ongoing debate on the definition of death (or life) among legislative bodies, as well as a lack of unifying guiding protocols or policies. This finding also highlights the potential challenges in applying traditional CPR termination criteria in resource-limited settings where confirmation of brain death through advanced imaging or electroencephalogram (EEG) may not be readily available.

The termination of resuscitative efforts in pediatric cardiopulmonary arrest is a complex decision that involves multiple factors, both medical and non-medical [18]. The decision is particularly challenging due to the emotional and ethical considerations involved, as well as the variability in the potential for recovery among pediatric patients. "There is no evidence to recommend a specific time period after which further resuscitation efforts are futile" [19]. However, pediatric CPR duration ranging from 10 to 20 minutes in neonates and 30 minutes (two 15-minute cycles plus asystole) in older children have been considered futile [20,21]. Further research is warranted to explore the ethical considerations and practical realities surrounding CPR termination practices in Sudanese pediatric contexts, particularly when brain death signs are present but cardiac activity persists.

From the author's experience, the most critical implications of such prolonged CPR practices are the strain on already scarce resources and stretched thin system; more specifically, the diversion of care from other more needy and prognostically promising patients. This typically involves utilizing medical staff to perform prolonged CPR, which leads to exhaustion and reduced availability to care for other patients.

Survey participants identified several significant barriers hindering the termination of prolonged, ultra-prolonged, and extreme-duration CPR. The most frequently cited obstacles were ethical concerns (84%) and the lack of clear protocols (83%) for making these decisions. These findings highlight the critical and urgent need for open discussions and the development of evidence-based guidelines that address the ethical complexities surrounding termination of CPR while considering the Sudanese cultural context. Furthermore, uncertainty about patient prognosis (66%) emerged as a major barrier, suggesting a need for appropriately designed studies addressing the association between the duration of CPR and survival and neurological outcomes in the local context of resources.

Thematic analysis of the open-ended question revealed several key areas surrounding prolonged CPR practices in Sudanese pediatric settings. A critical need for clear protocols and policies to guide decision-making around stopping CPR emerged as the most prominent theme. Devising a standardized approach has the potential for improved patient outcomes and reduced staff burden. The discussion around futility of prolonged CPR and the emotional toll on medical staff underscores the need for clear ethical frameworks for decision-making. Limited access to mechanical ventilation and post-resuscitation critical care were identified as significant barriers to successful resuscitation efforts. Critical care services are known to suffer the most in situations of resource limitation, given their resource-demanding structure. Sudan, in the midst of an armed conflict, is no exception. This necessitates a pragmatic approach, tailoring both practices and the research informing these practices to the unique local circumstances.

This study has some limitations. The reliance on self-reported data could result in recall bias, particularly concerning the details of CPR durations and outcomes, emphasizing the need for a more controlled prospective observational design. Another notable limitation is the use of purposive sampling for participant recruitment. Given the challenges of accessing a dispersed population in this context, this method was deemed the most practical. The author employed professional networks and WhatsApp groups as a pragmatic solution to enhance participation despite these constraints.

## Conclusions

This study revealed a relatively high frequency of ultra-prolonged and extreme-duration CPR in Sudanese pediatric settings. Termination decisions focused on cardiac activity becoming clinically undetectable rather than brain death signs or

specific cutoff CPR durations. The absence of a clear protocol on when to terminate CPR seems to contribute the most to this phenomenon. Developing standardized guidelines for prolonged CPR termination, addressing ethical and cultural/religious considerations given the peculiarity of the local setting, practices, and resources seems crucial for improving pediatric CPR outcomes in Sudan. Further research on patient outcomes after prolonged CPR in this context is warranted.

## Supporting information

**S1 File. Thematic analysis of the open-ended question.**
(PDF)

**S2 File. Dataset.**
(XLSX)

## Author contributions

**Conceptualization:** Mohammed Abdulrahman Alhassan.

**Data curation:** Mohammed Abdulrahman Alhassan.

**Formal analysis:** Mohammed Abdulrahman Alhassan.

**Funding acquisition:** Mohammed Abdulrahman Alhassan.

**Investigation:** Mohammed Abdulrahman Alhassan.

**Methodology:** Mohammed Abdulrahman Alhassan.

**Project administration:** Mohammed Abdulrahman Alhassan.

**Resources:** Mohammed Abdulrahman Alhassan.

**Visualization:** Mohammed Abdulrahman Alhassan.

**Writing – original draft:** Mohammed Abdulrahman Alhassan.

**Writing – review & editing:** Mohammed Abdulrahman Alhassan.

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
