## [Decision Letter · Decision Letter 0]

8 Jun 2025

PONE-D-24-28251When to Stop: Understanding the Landscape of Extreme-Duration Cardiopulmonary Resuscitation Practices among Pediatricians in SudanPLOS ONE

Dear Dr. Alhassan,

Thank you for submitting your manuscript to PLOS ONE. After careful consideration, we feel that it has merit but does not fully meet PLOS ONE’s publication criteria as it currently stands. Therefore, we invite you to submit a revised version of the manuscript that addresses the points raised during the review process.

We look forward to receiving your revised manuscript.

Kind regards,

Chiara Lazzeri

Academic Editor

PLOS ONE

Journal Requirements:

“The authors extend their appreciation to Prince Sattam bin Abdulaziz University for funding this research work through the project number (PSAU/2024/03/29191).”

Additional Editor Comments:

The population included in the study is quite peculiar. However major issue can be raised, mainly methodologic issues. Please provide a detailed reply to the reviewer's queries.

Reviewers' comments:

Reviewer's Responses to Questions

**Comments to the Author**

1. Is the manuscript technically sound, and do the data support the conclusions?

Reviewer #1: Partly

2. Has the statistical analysis been performed appropriately and rigorously? 

Reviewer #1: I Don't Know

3. Have the authors made all data underlying the findings in their manuscript fully available?

Reviewer #1: Yes

4. Is the manuscript presented in an intelligible fashion and written in standard English?

Reviewer #1: Yes

5. Review Comments to the Author

Reviewer #1: This may be an important study, but I have concerns about the Methodology of the study. My comments are:

Introduction:

‘The outcomes of prolonged CPR and the decisions surrounding when (and if) to discontinue CPR attempts remain subjects of further study, as well as academic and legislative debate.’

Can you provide more information on the academic and legislative debate and provide references if appropriate?

In this study, we aim to investigate the prevalence of prolonged (> 30 minutes- PCPR), ultra-prolonged (> 2 hours- UPCPR), and extreme-duration (>12 hours- EDCPR) BMV or CPR

I do not think you can calculate the prevalence using the method you have adopted.

Methods section:

How was the questionnaire developed? Was it validated and pre-tested? How was it translated from English to Arabic?

I am not aware about ethical approval in Sudan but as far as I know an institution in one country cannot provide ethical approval for research in another country. An institution in the country where the research was conducted should approve the study. If I am incorrect do let me know and please provide evidence.

How many paediatricians are currently working in Sudan?

There is only one author in this study so the use of the word ‘we’ may not be appropriate.

The Methodology may need to be explained better. How exactly was thematic analysis carried out?

I do not think figure 1 adds much useful information.

The study may have several limitations of Methodology as mentioned.

I am unable to recommend publication in the present form but would like to provide the author with a chance to revise.

6. PLOS authors have the option to publish the peer review history of their article (what does this mean? ). If published, this will include your full peer review and any attached files.

**Do you want your identity to be public for this peer review?** For information about this choice, including consent withdrawal, please see our Privacy Policy .

Reviewer #1: No

---

## [Author Response · Author response to Decision Letter 1]

26 Jun 2025

Original comments of the reviewer Reply by the author Changes done to manuscript

This may be an important study, but I have concerns about the Methodology of the study.

I thank the reviewer for the invaluable comments. They helped improve the manuscript substantially.

‘The outcomes of prolonged CPR and the decisions surrounding when (and if) to discontinue CPR attempts remain subjects of further study, as well as academic and legislative debate.’

Can you provide more information on the academic and legislative debate and provide references if appropriate?

I have added a brief paragraph addressing this important point.

Thank you. Introduction: “In high-resource settings, prolonged CPR in children is fraught with medical, ethical, and legal controversy. Studies show that extended resuscitation is usually associated with poor outcomes in pediatric cardiac arrest (8), yet deciding when to stop is extremely difficult. Ethically and legally there is strong pressure to “do everything” for a child – in fact, many jurisdictions exclude pediatrics from termination-of-resuscitation protocols, meaning providers must typically continue CPR until hospital arrival unless death is certain (9). This reluctance to cease efforts stems from outcome uncertainty: rare cases of survival after lengthy CPR make it hard to set firm cutoffs, so guidelines often emphasize case-by-case judgment and even advise involving the family in termination decisions once resuscitation has exceeded ~30 minutes with no response (9). In low-resource settings such as Sudan, these debates take on an additional significance as limited staffing, infrastructure, and supplies add a pragmatic layer. Local pediatric guidelines acknowledge that even basic equipment and drugs may be unavailable – standard CPR protocols have to be modified to fit reality (10). Consequently, discussions focus on how to balance ideal care with feasibility, and international experts have highlighted the need for context-specific resuscitation protocols, noting that many “international” CPR guidelines (designed for high-resource contexts) are often not applicable in resource-limited environments (11).”

In this study, we aim to investigate the prevalence of prolonged (> 30 minutes- PCPR), ultra-prolonged (> 2 hours- UPCPR), and extreme-duration (>12 hours- EDCPR) BMV or CPR

I do not think you can calculate the prevalence using the method you have adopted.

You are absolutely right.

I have changed the statement. Now “experience” is used instead of “prevalence”. “In this study, the author aims to investigate the pediatricians’ experiences with prolonged (> 30 minutes- PCPR), ultra-prolonged (> 2 hours- UPCPR), and extreme-duration (>12 hours- EDCPR) BMV or CPR.”

How was the questionnaire developed? Was it validated and pre-tested? How was it translated from English to Arabic?

I have added relevant statements addressing these concerns under the subsection “Questionnaire and data collection”. “It was developed by the author based on relevant literature, personal insights, and expert opinion to capture information on the frequency and circumstances of prolonged CPR episodes, decision-making processes, and barriers to terminating CPR.”

“A forward–backward translation process was used to ensure linguistic and conceptual consistency. Both versions were pilot-tested with five pediatricians, and only minor wording adjustments were made to enhance clarity.

Since the questionnaire did not involve scales or psychometric constructs, formal construct or criterion validity were not required. However, the questionnaire underwent expert review to ensure face and content validity. Feedback was obtained from pediatricians familiar with CPR practices to confirm that the items were clear, relevant, and aligned with the study objectives. Minor modifications were made accordingly.”

I am not aware about ethical approval in Sudan but as far as I know an institution in one country cannot provide ethical approval for research in another country. An institution in the country where the research was conducted should approve the study. If I am incorrect do let me know and please provide evidence.

Thank you for your comment. While it’s true that institutional review boards (IRBs) typically have jurisdiction limited to their own country, peer-reviewed ethical guidance supports the use of a single IRB for multinational, minimal-risk, web-based surveys among professionals—without requiring separate IRB approval in each recruiting country.

• Internet-based studies often fall under the purview of the coordinating institution’s IRB, given the global reach of online research. Ethics committees’ jurisdiction is geographically bounded, and it is common practice for anonymous, low-risk online surveys to proceed with approval from only one IRB.

• Local approvals become necessary only if the study involves physical interventions, collection of sensitive personal data.

Kindly review the following supporting authoritative reference:

https://doi.org/10.1016/j.invent.2021.100487

Here is a relevant quote from it: “it seems obvious for researchers to approach ethical review committees of participating institutes in international studies when physical activities are involved, such as providing study drugs or taking a blood sample. However, for international internet-based studies that take place solely online, it seems less obvious to apply for ethical review procedures in multiple countries.”

How many paediatricians are currently working in Sudan?

I have added the following statement to answer the reviewer’s question and address this point: “According to a 2010 report, Sudan had only 212 qualified pediatricians—with 112 being female—registered by that time (12). There's no more recent official national count available in the public domain” “According to a 2010 report, Sudan had only 212 qualified pediatricians—with 112 being female—registered by that time (12). There's no more recent official national count available in the public domain.”

There is only one author in this study so the use of the word ‘we’ may not be appropriate.

I have changed all the instances where “we” is used to “the author” instead.

Thank you. “the author” was used instead of “we”.

How exactly was thematic analysis carried out?

I have added a subsection to the methods section to address this.

Thank you. Thematic analysis

Open-ended responses were analyzed using thematic analysis. The author followed Braun and Clarke’s six-phase approach (13), beginning with familiarization with the data and generation of initial codes. These codes were then organized into overarching themes. Coding was conducted manually by the author, and emerging categories were refined iteratively. Responses in both English and Arabic were analyzed together after forward–backward translation. The final themes were derived inductively from the data and reflected recurring patterns of opinion and experience regarding prolonged CPR in pediatric settings. Minor language corrections were made to improve clarity without altering the original meaning.

I do not think figure 1 adds much useful information.

Thank you for the valuable feedback. I respectfully believe that Figure 1 offers a clear and immediate visual representation of where the majority of the data lie—something that cannot be easily conveyed through text alone. I would be grateful if the reviewer would consider retaining Figure 1 in the manuscript.

---

## [Editor Report · Decision Letter 1]

6 Jul 2025

When to Stop: Understanding the Landscape of Extreme-Duration Cardiopulmonary Resuscitation Practices among Pediatricians in Sudan

PONE-D-24-28251R1

Dear Dr. Alhassan,

We’re pleased to inform you that your manuscript has been judged scientifically suitable for publication and will be formally accepted for publication once it meets all outstanding technical requirements.

Kind regards,

Chiara Lazzeri

Academic Editor

PLOS ONE